# Exploring the Coordination and Spatial–Temporal Characteristics of the Tourism–Economy–Environment Development in the Pearl River Delta Urban Agglomeration, China

**DOI:** 10.3390/ijerph20031981

**Published:** 2023-01-21

**Authors:** Xueru Pang, Yuquan Zhou, Yiting Zhu, Chunshan Zhou

**Affiliations:** 1Key Laboratory of Sustainable Development of Xinjiang’s Historical and Cultural Tourism, Xinjiang University, Urumqi 830046, China; 2Department of Urban Planning and Spatial Analysis, Sol Price School of Public Policy, University of Southern California, Los Angeles, CA 90089, USA; 3School of Geography and Planning, Sun Yat-sen University, Guangzhou 510275, China

**Keywords:** tourism industry, economic development, eco-environment, coupling coordination degree, geo-detector model, the Pearl River Delta urban agglomeration

## Abstract

The rise of mass tourism has encouraged rapid economic growth; meanwhile, the eco-environmental system has come under increasing pressure. To achieve sustainable development, it is critical to deeply explore the relationship and evolution characteristics between three subsystems: tourism, the economy, and the eco-environment. This study aims to develop a more comprehensive indicator system for evaluating the coupling coordination degree (CCD) of the tourism–economy–environment (TEE) system using statistical data from nine cities in the Pearl River Delta (PRD) urban agglomeration from 2010 to 2019. We investigated the spatial–temporal evolution characteristics and driving forces of the TEE system in the PRD using the CCD model and the geo-detector model. The research results show the following: (1) The comprehensive benefits of the TEE system have increased steadily over the past 10 years, whereas the benefits of the eco-environment subsystem have fluctuated and been relatively unstable. (2) Spatially, in terms of tourism development, the eastern regions of the PRD are more developed than the western regions, and the regions with the greatest tourism benefits have gradually shifted to the northeastern regions of the PRD. Economic development presented an imbalanced but relatively stable spatial pattern. Guangzhou and Shenzhen have been the two most economically developed cities over the past 10 years. The eco-environment development has fluctuated over time, revealing a spatial pattern of cities with low environmental benefits in the center and cities with high eco-environmental benefits in the surrounding regions. (3) The PRD’s TEE system has become more integrated, moving from moderate disorder to a model of high-quality coordinated development, demonstrating a spatial pattern in which the cities of high development coordination are located near the Pearl River Estuary, and the coordination decreases the further away they are from the estuary. (4) The major driving factors of heterogeneous TEE coordination development include eco-environment protection, opening-up policies, education investment, technological innovation level, and the regional economic development level. The results are expected to effectively promote economic, tourism, and environmental improvement in the PRD, as well as to provide policy recommendations for coordinated TEE development in other similar urban agglomerations.

## 1. Introduction

As a comprehensive industry including food, housing, transportation, travel, shopping, and entertainment, tourism has become the third largest export industry in the world [1]. In 2019, the total number of global tourists reached 12.31 billion, an increase of 4.6% over the previous year. In addition, global tourism revenue reached USD 5.8 trillion in 2019, equivalent to 6.7% of the global GDP. Since the 40 years of reform and opening up, the tourism industry has experienced rapid development in China, and it has maintained a faster growth rate than the world tourism industry and China’s national economy. China is ranked the world’s third most popular tourist destination [2]. The United Nations World Tourism Organization shows that China’s tourism promotes economic development, increases employment, reduces poverty, and provides Chinese wisdom for the sustainable growth of world tourism. The rapid expansion of tourism industry drives the development of other social industries, thereby promoting socioeconomic development. In addition, tourism development is also consistent with the 17 United Nations Sustainable Development Goals (SDGs), which aim to solve major challenges being closely related to economic, social and eco-environmental issues and take the path of sustainable development [3]. Notably, 3 of the 17 SDGs are directly related to the tourism industry, including SDG 8, SDG 12, and SDG 14 [4]. Therefore, tourism development contributes to solving the constraints of the eco-environment on socioeconomic development.

Compared with other industries, tourism is characterized by low levels of consumption and environmental pollution. In the early 1970s, tourism was regarded as a “smokeless industry”, which mainly relied on developing natural and cultural resources as attractions for visitors [5]. However, with the development of tourism activities, people gradually have discovered that tourism is not a smokeless industry. While tourism activities bring economic benefits, the excessive pursuit of economic benefits has also resulted in serious problems in tourist destinations, such as ecological imbalance, pollution of the environment, and biodiversity loss. China’s economy is shifting from high-speed growth to a high-quality development stage, changing from traditionally focusing on economic growth to the coordinated development of social, economic, ecological, and other aspects. Therefore, how to coordinate the development of economic growth, tourism, and the eco-environment has become an important issue concerned by academia and government.

The tourism–economy–environment (TEE) system is an open construct with rich connotations, a complex structure, and coupling characteristics [6]. First, as a sunrise industry and comprehensive industry, tourism has become one of the fastest growing and rapidly developing industries in the 21st century. The rapid growth of tourism industry has driven the boom of the catering industry, commerce, the hotel industry, the transportation industry, and other related industries. It has also optimized the industrial structure, increased employment opportunities, and promoted economic growth [7,8]. Meanwhile, stable economic development has provided infrastructure and construction funds for tourism. Nevertheless, excessive and rapid development related to tourism damages the local environment and harms tourism economic growth, thus inhibiting tourism development in the long run [9]. Second, the eco-environment provides natural resources for tourism development. A sound eco-environment increases the attractiveness of the destination and improves tourist satisfaction. While the worsening of eco-environment leads to a decrease in environmental quality in tourist destinations, threatening the healthy development of tourism [10]. Finally, economic growth and the ecological environment are also interrelated. A sound eco-environment provides basic material guarantees for economic development, such as clean water, fresh air, and abundant energy. Economic development can provide financial guarantees and technical support for protection of the eco-environment. However, blindly pursuing economic interests results in the serious damage to eco-environment and exceeds the corresponding threshold [11,12], which affects the sustainable way of tourism development. In other words, there is a coupling relationship between the tourism industry, economic development, and eco-environment subsystems, as shown in Figure 1. Thus, it is theoretically and practically of great value to understand the coupled coordination relationship in the three subsystems, which not only provides a scientific basis for the sustainable and healthy development of tourism industry but is also conducive to coordinating socioeconomic growth and environmental improvement to achieve sustainable regional development.

As a crucial aspect of sustainable development, the relationship between tourism, the economy, and the eco-environment has attracted increasing scholarly attention. Existing researches have mainly focused on three issues. The first concerns the correlations between economic growth and the eco-environment; research on this topic was carried out early, with increasingly rich research results. The most typical is the environmental Kuznets curve (EKC) proposed by Krueger in the 1950s, which reveals the inverted U-shaped relationship between economic growth and environmental degradation [13]; that is, with economic growth and the increase in per capita income, environmental quality deteriorates initially and improves after economic increase reaches a certain level. The effectiveness of the EKC has been studied and tested extensively [14,15,16], with some scholars questioning its validity. For example, Dinda deemed that the relation of eco-environmental pressure to economic income originally exhibits an inverted-U curve; however, beyond a certain income level, the relations between them becomes an N shape [17]. With regard to the correlations between the economy and the eco-environment, fruitful research methods were utilized, including the coupling coordination degree (CCD) model [18], weighted TOPSIS [19], geospatial analysis [20], the dynamic deviation maximization (DM) method [21], the logistic model [22], and the gray system model [23]. Of these, the CCD model has been particularly widely used [24,25] because it is more suitable for studying each subsystem’s interactive relationships and degree of coordination. In research on the coordinated development of the environment and economy in the Yangtze River Delta of China, scholars have measured the CCD and explored its spatial–temporal evolution as well as influencing factors; they found that GDP and the annual average concentration of PM2.5 were the factors that most influenced the CCD [26].

The second issue concerns the correlations between the tourism industry and economic growth. As a tertiary industry, tourism has driven other related industries’ development and has become an essential factor in maintaining sustainable economic development. The research scale involves the nation [27], province [28], and city [29] of which medium-macro scales are dominant. In addition, the research methods include the cointegration model [30], the social accounting matrix (SAM) model [31], cluster analysis [32], and the computable general equilibrium (CGE) model [33]. The research contents revolve around the influence of the economy on tourism [34], the influence of tourism on the economy [35], and the interaction between them [36]. With the concepts of green economy, low-carbon development, and sustainable development being put forward, research aimed at the transformation of tourism into a green economy and sustainable tourism development. Law et al. proposed a framework for a green economy transformation of tourism and developed a Green Growth 2050 Roadmap for Bali’s tourism [37]. In addition, some scholars proposed six significant interdisciplinary elements of sustainable tourism, including green transportation, green energy, green buildings, green agriculture, green infrastructure, and smart technologies [38].

The third point concerns the interaction between the tourism industry and the eco-environment, which is very important. Tourism activities affect the local ecosystem by placing pressure on the local water, air, land, forest, and other relevant ecosystems. SDGs 14 and 15 (aquatic and terrestrial ecosystems) both emphasize the importance of clean ecosystems for sustainable development [39]. Scholars increasingly research the prominent contradictory relations between the tourism industry and the eco-environment. Existing studies have mainly focused on the negative effect of tourism on the eco-environment or the deterioration of the eco-environment restricting tourism. In the case of Kosciuszko National Park in Australia, it was found that tourism infrastructure had an impact on the diversity of exotic plants in protected areas [40]. The assessment of tourism’s total global resource use shows that maintaining the global tourism system needs rapid growth of resource inputs, which in turn have threatened the eco-environment [41]. Some scholars have demonstrated the influence of climate change on the tourism industry through case studies [42,43]. If tourism resources are reasonably developed, the economic benefits created by the tourism industry will provide more financial resource and technical support for ecological protection. Using the fully modified ordinary least squares approach, scholars have discussed the relationship between tourism and environmental pollution in the three lower middle-income Southeast Asian economies and found that tourism has adversely affected the eco-environment for Indonesia and the Philippines, while tourism has improved the environmental quality of Vietnam [44]. Research has recently focused on the interrelation mechanism between tourism and the eco-environment [45]. For example, the spatial–temporal evolution and the reason for the changes in the decoupling relationship between China’s tourism growth and eco-environment pressure from 2009 to 2018 were explored [46].

The above literature shows the gradual shift from initial research on the mutual relations between the eco-environment and the economy to the interrelation between tourism growth and economic development or the eco-environment. Most studies focus on measuring the relationship between the two, while relatively few studies combine tourism, economy, and environment for analysis. In the case of Shandong Province, Li et al. made use of the entropy method and the CCD model to quantitatively study the coupled coordination relationship and development law of tourism, the environment, and regional economic systems [47]. Zhang et al. discussed the characteristics and influencing factors of the CCD of the tourism–economy–environment system, using the data of 12 provinces and cities in western China [48]. In addition, most existing studies take multiple provinces [49] or a single province [50] as the research object, calculate the CCD, and analyze the time variation characteristics. However, few studies take urban agglomeration as the research object to analyze the spatial–temporal characteristics and influencing factors of the CCD, combining time and space dimensions. 

Urban agglomeration refers to the highly developed spatial form of integrated cities [51]. It is the most dynamic and potential growth pole in the countries’ or regions’ economic development [52]. The conflict among economic growth, tourism development, and environmental protection in urban agglomeration has become a pressing concern over the years. Therefore, focusing on the sustainable development of urban agglomeration is crucial for enhancing national competitiveness. China is gradually building urban agglomerations, and has approved 10 national level urban agglomerations. Since the reform and opening up, the Pearl River Delta (PRD) urban agglomeration has become an engine of national economic growth. Thanks to its advantaged location and the support of national preferential policies, the PRD has attracted a large amount of foreign capital and developed its economy rapidly. Under the current strategic background of the “the Belt and Road” and the construction of the Guangdong–Hong Kong–Macao Greater Bay Area, the PRD urban agglomeration ushers in new development opportunities and becomes an important part of the “the Belt and Road”, especially the 21st Century Maritime Silk Road. With rapid industrialization and urbanization, the PRD urban agglomeration is facing a huge disturbance of weak eco-environment bearing capacity and environmental pollution. The economic development stage, industrial structure, and ecological conditions of each prefecture level city are different; however, few studies have explored the balance of the three systems in the PRD urban agglomeration. Under the background of aiming towards high-quality and sustainable development, the PRD urban agglomeration is faced with the severe challenge of coordinating the relationship among economic growth, tourism development, and eco-environmental protection. Meanwhile, establishing a reasonable and comprehensive evaluation indicator system is the theoretical prerequisite for studying the coupled coordination relationship and evolution characteristics of the TEE system in the PRD, as well as the practical needs of building a world-class urban agglomeration and tourism destination in the Guangdong–Hong Kong–Macao Greater Bay Area.

Therefore, this study attempts to select the representative PRD urban agglomeration as the research area and constructs an indicator system of three subsystems. The CCD model and geo-detector model are used to evaluate the spatial–temporal evolution characteristics and influencing factors of the CCD of TEE system in the PRD urban agglomeration from 2010 to 2019. Considering the impact of the COVID-19 pandemic, all sectors, especially the tourism industry, have suffered a severe and unprecedented impact. In order to prevent the pandemic-related abnormal data in 2020 biasing our analysis, we selected 2010–2019 as our study period. This study aims to give some practical suggestions for the region’s coordinated development and provide a reference for the high-quality development of other urban agglomerations.

## 2. Data Sources

### 2.1. Study Area

The PRD urban agglomeration is located in the south-central region of Guangdong Province, facing the South China Sea and adjacent to Hong Kong and Macao. It is one of the most dynamic economic regions in the Asia Pacific region. The PRD urban agglomeration includes nine cities: Guangzhou, Shenzhen, Foshan, Dongguan, Zhongshan, Zhuhai, Jiangmen, Zhaoqing, and Huizhou, ranging over a total area of 55,368.7 km^2^. It is one of the largest urban agglomerations in the world and one of the three regions with the strongest innovation capability, the largest population density and the highest comprehensive strength in China (Figure 2). In 2019, the permanent population of the PRD was 64.469 million, accounting for 4.6% of China’s total population, while the GDP of the PRD reached RMB 8.69 trillion, accounting for 8.8% of China’s GDP. The foreign exchange income of international tourism was USD 19.217 billion, accounting for 14.64% of China. Hence, the PRD is one of the regions with the most developed tourism economy in China. Compared with the Yangtze River Delta, Beijing–Tianjin–Hebei and other urban agglomerations, the PRD urban agglomeration formed an export-oriented economy by virtue of its regional advantages and preferential policies, quickly became the “world factory”, and therefore, experienced the most rapid urbanization since the reform and opening-up. Currently, the PRD, together with the Hong Kong Special Administrative Region (SAR) and the Macao SAR, has established the Guangdong–Hong Kong–Macao Greater Bay Area, one of the world’s four major harbor areas. Compared with the other three world-class bay areas, the Guangdong–Hong Kong–Macao Greater Bay Area has advantages in area, population, GDP, and scale and has huge development potential in per capita GDP, industrial structure and innovation ability. Due to limited data and inconsistent currency in Hong Kong and Macau SAR, this study chooses the PRD urban agglomeration as the research object, which is a major and vital part of the Guangdong–Hong Kong–Macao Greater Bay Area. In addition, the expansion of economic scale and the rapid development of tourism industry have led to environmental pollution. The TEE system of the PRD urban agglomeration is complex in structure and present strong coupling interactions, making it an area worth exploring.

### 2.2. Research Framework

To research the coupled coordination relationship and evolution characteristics of the TEE system of the PRD urban agglomeration from 2010 to 2019, this study first constructs the indicators of the three subsystems and calculates the indicator weights. Then, based on the CCD model, the benefit index of the three subsystems and comprehensive benefit evaluation index of the TEE system and the CCD are analyzed from spatial and temporal dimensions. Additionally, the influencing factors of CCD are determined by the geo-detector model. Finally, we present a discussion and put forward relevant suggestions according to the results. The specific study framework and implementation steps are as follows (Figure 3). 

### 2.3. Data Source

The data used in this study are from the Guangdong statistical yearbook, China urban construction statistical yearbook, and China city statistical yearbook from 2010 to 2019. Some of the data are from the China tourism statistical yearbook, the statistical bulletin of national economic and social development of Guangdong Province, the yearbook of Guangdong tourism, the statistical yearbook and environmental quality bulletin of the nine cities, as well as some government authoritative websites such as the Guangdong provincial statistics information network and the department of culture and tourism of Guangdong Province. The missing values are predicted and supplemented by the linear trend of the adjacent points of the corresponding indicators with the help of SPSS 27.0 software. 

### 2.4. Construction of the Indicator System

Based on the principles of being representative, integral, scientific, operational, and comparative, we combine the existing relevant research results [47,50,53] and the specific situation of the PRD to objectively build an indicator system that fully reflects the coupled coordination relationship of the TEE system (as shown in Table 1). The system consists of three layers: the tourism industry system, economic development system, and eco-environment system, including 28 indicators. Among them, the tourism industry system consists of 3 primary indicators: tourism economic benefits, tourism market scale, and tourism reception facilities, including 8 secondary indicators; the economic development system consists of 3 primary indicators: economic development scale, economic development structure, and economic development level, including 10 secondary indicators; and the eco-environment system consists of 3 primary indicators: ecological construction, environmental pollution, and environmental pollution treatment, including 10 secondary indicators. Notably, Y_21_ measures the impact of economic non-agricultural transformation, and a low value indicates an upgrade of industrial and economic structure. Considering the actual industrial structure of the PRD in recent years, the proportion of primary industry is declining, as well as in the previous literature [23,50], indicating it a negative indicator of economic development.

### 2.5. Research Methods

#### 2.5.1. Entropy Method

To determine indicators’ weight, the entropy weight method, which is more objective than other methods, was adopted. The specific calculation steps are shown as follows: 

Step 1: Standardization of indicator data:

Consideration of the difference in dimensions and attributes of the indicators, dimensionless processing is required to ensure the validity of the data. In this paper, the range method is used for data standardization, and the equations are as follows: (1)$$\mathrm{Positive}\;\mathrm{indicators}:X_{ijk}=\frac{x_{ij}-\min\{x_{j}\}}{\max\{x_{j}\}-\min\{x_{j}\}}$$
(2)$$\mathrm{Negative}\;\mathrm{indicators}:X_{ijk}=\frac{\max\{x_{j}\}-x_{ij}}{\max\{x_{j}\}-\min\{x_{j}\}}$$
where $X_{ijk}$ represents the standardized values of the jth indicator in the kth city in the ith year, and $\max\{x_{j}\}$ and $\min\{x_{j}\}$ represent the maximum and minimum values, respectively.

Step 2: Calculation of the indicator proportions:(3)$$p_{ijk}=\frac{X_{ijk}}{\sum_{i=1}^{n}X_{ijk}}$$
where n denotes years, and $p_{ijk}$ represents the proportion of the jth indicator in the kth city in the ith year.

Step 3: Calculation of the entropy value of the indicator:(4)$$E_{j}=-\ln{\left(n\right)}^{-1}\sum_{i=1}^{n}p_{ijk}\ln(p_{ijk})$$

Step 4: Determining the indicator weight:(5)$$W_{j}=\left(1-E_{j}\right)/\sum_{i=1}^{m}\left(1-E_{j}\right)$$
where m is the number of indicators, $E_{j}$ denotes the information entropy of the jth indicator, and $W_{j}$ denotes the weight of the jth indicator.

#### 2.5.2. The CCD Model and Division Standard

Comprehensive evaluation index

The linear weighting method was used to calculate the coordination status of the TEE system in the PRD from 2010 to 2019. The benefit functions of the three subsystems of the tourism industry, economic development, and eco-environment are as follows:(6)$$F(x)=\sum_{i=1}^{m}W_{Ai}X_{Ai}$$
(7)$$G(y)=\sum_{i=1}^{m}W_{Bi}X_{Bi}$$
(8)$$H(z)=\sum_{i=1}^{m}W_{Ci}X_{Ci}$$
(9)T=αF(x)+βG(y)+rH(z)
where $W_{Ai}$, $W_{Bi}$, and $W_{Ci}$ are the weights of the indicators in the three subsystems; $X_{Ai}$, $X_{Bi}$, and $X_{Ci}$ are the standardized values after dimensionless processing of each subsystem; F(x), G(y), and H(z) represent the benefit index of the three subsystems of the TEE; T is a comprehensive benefit evaluation index for TEE system that represents the overall development level; and α, β, and r represent the weights of the three subsystems. Referring to previous studies [47,50], in this study, we took α = 0.2, β = 0.4, and r = 0.4, combining the interrelationship between the tourism industry, economic growth, and eco-environment in the PRD.

2.The CCD model

The concept of coupling is derived from physics, which denotes the phenomenon that two or more systems influence each other by interacting. The CCD model has been widely used in the research of economy, ecology, resources, urbanization, and other fields. The calculation model is:(10)$$C={\left\{\frac{F(x)\times G(y)\times H(z)}{{\left[\frac{F(x)+G(y)+H(z)}{3}\right]}^{3}}\right\}}^{\frac{1}{3}}$$
where C denotes the CCD of the TEE system and 0≤C≤1. When C=1, it indicates that the three subsystems are in the best coupling state; when C=0, it indicates that the three subsystems are irrelevant and disorderly; when the value of C is larger, the three subsystems interact strongly, and the degree of couplings is higher, and vice versa.

The coupling degree only describes the degree of interaction among the subsystems and does not fully represent the coordinated development state of the subsystems. Hence, there is a need to further analyze the coordination degree of TEE with the help of the CCD model. The formula is as follows:(11)$$D=\sqrt{C\times T}$$
where D is the CCD of the three subsystems (0≤D≤1). The closer the value of D is to 1, the better the coupling coordination state of the three subsystems.

Referring to the previous research [47,54], in this study, the CCD is classified as shown in Table 2.

#### 2.5.3. Geo-Detector Model

The geo-detector model is a method applied to analyze the spatial differentiation laws of geographical phenomena and reveal their influencing factors. It was first used to study the impact of geographical spatial factors on diseases and has been widely used in land use, geology, socioeconomic development, the ecological environment, and other fields. In this paper, factor detector was used to detect and analyze the driving factors of the CCD of the TEE system in the PRD. The model is as follows [55]:(12)$$P_{D,U}=1-\frac{1}{n\sigma^{2}{}_{U}}\sum_{i=1}^{L}n_{D,i}\sigma^{2}{}_{D,i}$$
where $P_{D,U}$ measures the influence of the factor D, whose value range is [0, 1]. The higher the value is, the greater the impact of the factor D on the spatial differentiation of CCD will be. $n_{D,i}$ and n are the number of samples in the subresearch area and the whole research area, respectively. L represents the number of categories. $\sigma^{2}{}_{D,i}$ and $\sigma^{2}{}_{U}$ represent the variance of CCD of the subresearch area and the whole study area, respectively.

With the consideration of the existing literature [52,56], the PRD’s development pattern, and the availability of data, this paper selected the total value of actual utilized foreign capital to represent the degree of opening up (X1), the proportion of education expenditure in fiscal expenditure to represent education investment (X2), GDP growth rate to represent regional economic development level (X3), proportion of second and tertiary industry in GDP to represent industrial structure (X4), number of A-level scenic spots to represent tourism resource endowment (X5), total capacity of sewage treatment to represent eco-environment protection (X6), year-end population to represent population size (X7), and number of ordinary colleges and universities to represent technological innovation level (X8), mainly exploring the influencing factors of the CCD of the TEE system in the PRD in eight dimensions (Table 3).

The data used by the geo-detector model include the dependent variable Y and the independent variable X, and the independent variable is the categorical data. With the assistance of ArcGIS software, the natural breakpoint classification method was used for spatial discretization, and the numerical data were converted into categorical data. In addition, ArcGIS software was used to grid the research area, create a fishing net of 8 km × 8 km, with a total of 858 grid points, and extract the corresponding X and Y in space.

## 3. Results

### 3.1. Comprehensive Benefit Index Analysis of the TEE System

#### 3.1.1. Temporal Analysis of the Benefit Index of Each Subsystem in the PRD

The weight coefficients of all indicators in the tourism industry, economic development and eco-environment systems in the PRD were obtained by using the entropy weight method, as shown in Table 1. Then, the benefit index of each subsystem and the comprehensive benefit evaluation index of TEE system in the PRD from 2010 to 2019 was calculated using the linear weighting method, as shown in Figure 4.

Figure 4 shows that the comprehensive benefit evaluation index of the TEE system in the PRD presented an upward trend from 2010 to 2019, rising from 0.174 to 0.822. The eco-environment system is relatively unstable. The eco-environmental benefits presented a small upward trend from 2010 to 2012 and then decreased to 0.489 in 2013. The eco-environmental benefit index showed a slightly inverted “U” trend from 2010 to 2013, showing a more significant inverted “U” shape after 2013. Therefore, the eco-environment development trend in the PRD generally conformed to a positive “U” environmental Kuznets curve. Eco-environmental pollution decreased with the progress of the economy and tourism during early development. However, after the inflection point, the level of eco-environment pollution increased with the regional economic growth. Both benefit indexes of the economic development and the tourism industry subsystem showed an upward trend, among which the economic development subsystem rose from 0.041 in 2010 to 0.916 in 2019, and the uptrend of the tourism industry subsystem was more significant, rising rapidly from 0.01 in 2010 to 0.985 in 2019. China put forward the “the Belt and Road” in 2013, the “Pearl River Delta Global Plan” in 2014, and the development strategy of the Guangdong–Hong Kong–Macao Greater Bay Area in 2017. Relevant policies have facilitated the tourism and economy development in the PRD. In general, the TEE system and its subsystems in the PRD mainly tended to rise from 2010 to 2019, except for the eco-environment system. The eco-environmental benefits in the PRD were not ideal, which was affected by urban land expansion, human activities, pollution emissions, natural disasters, and other factors. While developing the economy and tourism industry, optimizing the eco-environment will promote regional sustainable development.

#### 3.1.2. Spatial Analysis of the Benefit Index of Each Subsystem of Nine Cities in the PRD

The benefit index of the tourism industry, economic development and eco-environment subsystems of nine cities in the PRD from 2010 to 2019 were calculated according to Equations (6)–(9), and their line charts were drawn, as shown in Figures 5, 7 and 9. The three years 2010, 2014, and 2019 were chosen for use in ArcGIS to explore the spatial evolution characteristics, as shown in Figures 6, 8 and 10. The Jenks natural breaks was used to classify the benefit index of three subsystems and the TEE system in the nine cities in the PRD into five levels from low to high (low, low-medium, medium, upper-high, and high).

Tourism development in nine cities in the PRD generally increases, but there are significant regional differences (Figure 5 and Figure 6). Figure 5 shows that the tourism benefits of nine cities in the PRD can be divided into two development tiers. Guangzhou and Shenzhen are first-tier cities and their tourism development levels are far ahead of the other cities. The tourism benefits of these two cities were higher than 0.3 and showed an upward tendency from 2010 to 2019. The other seven cities are located in the second tier. The tourism benefits in these seven cities were relatively low during the study period, 0.2 or below. Although the tourism development of the second-tier cities shows an upward trend overall, the development level is far lower than that of the first-tier cities. The reason was that Guangzhou, as the central city in southern China and the most representative city of Lingnan culture area, not only has rich tourism resources and a long history, but also has excellent tourism facilities and services. Shenzhen’s tourism industry has strong marketing influence and tourism reception ability, attracting tourists with the charm of a modern city. However, the tourism development in the second-tier cities is still restricted by some factors. For example, Zhuhai is adjacent to Macao, with good air quality, livable climate, and beautiful coastal landscapes, and has been selected as one of the “Top 40 Tourist Resorts in China”, but there are problems such as imperfect public service facilities and tourism infrastructure. Dongguan has complete tourism development infrastructure, but lacks core cultural competitiveness and cultural heritage.

Figure 6 shows that the tourism development of Guangzhou and Shenzhen was at a high level in 2010, forming a dual-nuclei spatial structure. The Guangzhou–Shenzhen dual center model has driven the surrounding cities. In 2010, Zhuhai was at an upper-high level; Dongguan, Foshan, and Huizhou were at a middle level; Zhaoqing and Jiangmen were at a low-medium level; and Zhongshan was at a low level.

In 2014, Guangzhou and Shenzhen were still at relatively high levels. Dongguan, located in the eastern and central regions of the PRD, rose from a medium level to an upper-high level. Jiangmen, Foshan, and Zhongshan in the west of the PRD remained at the same level. Although Zhaoqing and Huizhou dropped by one level, respectively, the five levels of the tourism benefits in 2014 and 2019 increased compared with 2010, which indicated that the tourism development level in the PRD has improved.

In 2019, the tourism benefits of Guangzhou and Shenzhen still maintained leading positions, and Huizhou rose by two levels, which shows a trend of Guangzhou and Shenzhen as cores and the surrounding cities decreasing. The reason was that Huizhou has seized the new opportunity of the Greater Bay Area construction and development, improved tourism infrastructure, built leisure base of coastal tourism, and prioritized the development of leisure vacation tourism and rural tourism. Overall, the tourism benefits in the PRD have improved, which was related to macro policy support. The 12th Five-year Tourism Plan (2011–2015) of Guangdong Province proposed building the PRD into a metropolitan tourism area of Guangdong Province. The 13th Five-year Tourism Plan (2016–2020) of Guangdong Province comprehensively promoted all-for-one tourism and accelerated constructing world-class tourism and leisure destinations. These policies and measures have provided a broader development space for the tourism in Guangdong Province, especially the PRD. The increase in tourist arrivals, income, and A-level scenic spots was the direct reason for the improvement of tourism benefits of these cities. In addition, the benefit index of the tourism industry in the PRD showed was high in the east and low in the west, and the region with the greatest tourism benefits gradually shifted to the northeast regions of the PRD. These results also showed that the gap between high-value regions and low-value regions of tourism benefits has increased, and the difference between the eastern and western areas of the PRD is significant. Some regions have not maintained their original industrial development advantages in the process of development, resulting in relative regression.

The benefit index of the economic development system in the PRD is uneven in spatial distribution, but it presents a relatively stable spatial distribution pattern (Figure 7 and Figure 8). Figure 7 illustrates that the economic benefits of nine cities in the PRD showed an annual upward trend: Shenzhen ranked first, Guangzhou second, followed by Foshan, Dongguan, Zhuhai, Zhongshan, Huizhou, Jiangmen and Zhaoqing.

Figure 8 shows that Guangzhou and Shenzhen were high-level economic development regions in 2010. In 2014, the economic growth advantages of them were more significant than those of the surrounding cities in the PRD. The economic benefits of Jiangmen in 2019 increased by one level compared with 2014, and the value of each level of the economic development subsystem increased annually. The reason was that the relevant indicators of economic development have risen. For example, the proportion of tertiary industry in GDP increased from 42.88% in 2014 to 48.93% in 2019 for Jiangmen. Overall, the economic benefits have formed a high-value cluster with Guangzhou and Shenzhen as the center. As provincial capital cities and special economic zones, Guangzhou and Shenzhen have unique geographical conditions and market advantages, which give play to the siphoning effect. Accordingly, the proportion of tertiary industry in GDP, total export-import volume, GDP, per capita disposable income, and other indicators in Guangzhou and Shenzhen were far higher than those of other cities. The economic development levels of Foshan, Dongguan, and Zhuhai were also ranked in the forefront of the PRD, indicating that the closer the city is to the economic core area, the more significant the radiating effect is. In addition, the surrounding cities were low-value regions, among which Zhaoqing in the northwest, Jiangmen in the southwest, and Huizhou in the northeast were relatively backward in the PRD. The main reason is that the second and third industries started relatively late, and their development speed is relatively slow.

The benefit index of the eco-environment subsystem in the PRD is unstable, as evidenced by a spatial pattern of cities with low environmental benefits in the center and cities with high environmental benefits in the surrounding regions (Figure 9 and Figure 10).

Figure 9 shows that the benefit index of the eco-environment subsystem for most cities in the PRD had inflection points in 2013 and 2015, and after 2015, they showed a downward trend, indicating the problem of environmental degradation. By looking through the eco-environment subsystem indicators, we found that the common reasons for the decline of the eco-environmental benefits of each city were the increase in industrial “three wastes” emissions, and decline of green coverage rate in built-up areas and forest coverage. For example, the production of industrial solid wastes in Guangzhou increased from 4,596,300 tons in 2015 to 7,663,200 tons in 2019, and the green coverage rate in built-up areas in Zhuhai decreased from 58.11% in 2015 to 46.87% in 2019. In addition, Zhaoqing and Huizhou, which are relatively backward in terms of tourism and economic development, have always been at the lead in terms of eco-environmental benefits.

Figure 10 shows that the high-value regions of eco-environmental benefits were Zhaoqing and Huizhou in 2010, mainly because the industrial waste gas emission and the production of industrial solid wastes were low, and the forest coverage was relatively high in these two cities. In addition, Guangdong Province put forward the “Double Transfer” strategy (industry and labor migration) in 2008 and Zhaoqing and Huizhou actively introduced high-quality industrial projects. Considering the impact of terrain and geographical location, with mountains and hills, the urbanization expansion was relatively slow, and the ecological environment remained at a high level. However, due to industrial structure adjustment, the absorption of a large number of laborers, and the acceleration of urbanization, Foshan has faced serious eco-environment damage and environmental pollution problems. In 2014, Dongguan’s eco-environmental benefits declined by one level. In 2019, Shenzhen, Huizhou, and Zhongshan’s levels declined relatively, but the overall benefits were improved. Low or low-medium areas have been formed in the central PRD (Foshan, Zhongshan, Dongguan and Guangzhou), owing to large pollution emissions, low forest coverage, low terrain, a relatively closed surrounding environment, weak air mobility, and poor air self-purification capacity. In particular, Dongguan was affected by the highly polluting manufacturing industry and the “Three-plus-one” trading mix (custom manufacturing with materials, designs or samples supplied, and compensation trade).

Therefore, the eco-environmental benefits in the PRD shows a fluctuating trend on the whole, indicating that the PRD has experienced an ecological deficit and environmental pollution problems resulted from the pursuit of rapid economic growth and urbanization expansion. The city governments in the PRD are paying increasing attention to eco-environmental protection and are gradually transitioning from the original extensive development mode at the expense of a healthy environment to the high-quality development mode characterized by low energy consumption and low levels of pollution.

### 3.2. Spatial–Temporal Evolution of the CCD of the TEE System

#### 3.2.1. The Temporal Evolution of the CCD

The benefit index of the tourism industry, economic development, and eco-environment subsystems of nine cities in the PRD from 2010 to 2019 were entered into the coupling degree and CCD calculation formulas (Formulas (10) and (11)) to gain the coupling degree and CCD of the TEE system (Table 4 and Figure 11). 

From the perspective of time evolution, the coupling degree and CCD of the TEE system in the PRD gradually improved from 2010 to 2019. The speed of improvement markedly increased in the early stages and then slowed in its later stages. The CCD type gradually transitioned from moderate disorder to high-quality coordination. Taking 2013 as the transition period, the CCD of the TEE system can be divided into two stages: from 2010 to 2013, when it transitioned from moderate disorder, to approaching disorder and primary coordination; and from 2014 to 2019, when it transitioned from intermediate coordination, to well coordination and high-quality coordination. We found that the cobenefit of the PRD was weak before 2013, with weak interaction between subsystems. The CCD of the TEE system rose in 2013, with strengthened system interactions, changing from intermediate to well coordination. Due to the differentiation of development levels of the three subsystems, although the CCD improved, there was still a lag effect. There are three lag types: tourism lagging, economic lagging, and environment lagging. According to Table 3, the CCD of the TEE system belonged to the tourism lagging type from 2010 to 2015, indicating that the tourism development of the PRD was in the early stage with a low development level. From 2016 to 2017, it was of the economic lagging type, with good eco-environment quality but relatively slow economic development. After 2018, the environment lagging type emerged in response to the rapid growth of tourism industry and the economy, having a negative effect on the eco-environment. 

#### 3.2.2. The Spatial Evolution of the CCD

To further demonstrate the spatial distribution characteristics, this study calculated the CCD of the TEE system of nine cities in the PRD from 2010 to 2019 according to Equations (10) and (11), by drawing a line chart (Figure 12), and selecting 2010, 2014, and 2019 as three representative years to perform visual expression by using ArcGIS (Figure 13).

Overall, the CCD of the three subsystems in the PRD has gradually improved in the past decade. However, the development level of nine cities in the PRD was not balanced, and there was also regional differentiation in their CCD (Figure 12 and Figure 13). As shown in Figure 12, the CCD of the TEE system of the nine cities in the PRD fluctuated upward, and the coordination development level of the three subsystems in nine cities has made progress with improved coordination types, meaning that the CCD is developing in a positive direction. This change was formed by the continuous improvement of the economic development level, tourism and eco-environment, which reflected that the PRD has realized the optimization and upgrading of the industrial structure, and has strengthened environmental governance and ecological restoration.

As shown in Figure 13, from 2010 to 2019, there was significant regional differentiation in the CCD of the TEE system between the nine cities in the PRD, presenting a spatial distribution pattern of the cities near the Pearl River Estuary (Guang, Shen and Zhu) as the center and decreasing to both sides. In 2010, Guangzhou and Shenzhen were in the state of primary coordination, while the other seven cities were in a state of mild or approaching disorder. In 2014, the CCDs of nine cities in the PRD improved by one level. In 2019, the CCD of Jiangmen, Zhuhai, Dongguan, Guangzhou, and Shenzhen improved, among which Guangzhou and Shenzhen improved from moderate to well coordination, with more significant regional differentiation.

Specifically, Guangzhou and Shenzhen have a high level of CCD in the past decade, playing a central role in the development of the PRD urban agglomeration. Guangzhou, as the provincial capital in Guangdong Province and one of China’s national central cities, is an international trade center and integrated transportation hub, which not only becomes the central city of the PRD region but also the hub city of the Belt and Road. As China’s first special economic zone and the window of reform and opening up, Shenzhen has become one of the four central cities in the Guangdong–Hong Kong–Macao Greater Bay Area. Both cities have superior geographical locations, good economic conditions, developed transportation hubs and advanced science and technology. More importantly, the two cities adopted industrial transfer policies early to realize the transformation and upgrading of high-tech industries, and introduced a series of environmental governance policies to reduce pollution and carbon. For example, Shenzhen took the lead to establish the medium and long-term plan for low-carbon development of Shenzhen in 2012. Although Dongguan and Huizhou in the eastern areas of the PRD have undertaken the industrial transfer from Shenzhen and Hong Kong to promote economic development, their demographic dividend has weakened, so they need to improve their technological innovation capabilities to achieve environment-friendly development. However, the coordinated development of Zhaoqing, Foshan, Zhongshan, and Jiangmen in the western areas of the PRD is relatively slow, which is caused by slow industry upgrading, stagnation of technology, inconvenient transportation, and lacking of certain geographical advantages and relevant policy support. It can be seen that there are overall convergence and individual differentiation of CCD development in the PRD urban agglomeration.

### 3.3. Influencing Factors of CCD of the PRD’s TEE System

Taking the above eight influencing factors as detection factors, the three years 2010, 2014 and 2019 were selected to explore the differences in the effects of various factors on the spatiotemporal evolution of the CCD of the TEE system in the PRD using the geo-detector model. The results of factor detection (Table 5) showed that the explanatory power (q-value) of various influencing factors on CCD were ranked from strong to weak as follows: eco-environment protection (X6), degree of opening-up (X1), education investment (X2), technological innovation level (X8), regional economic development level (X3), tourism resource endowment (X5), population size (X7), and industrial structure (X4).

The closer q is to 1, the greater the explanatory power of the influencing factor to the spatiotemporal differentiation of the CCD. All q values in Table 5 are greater than 0.5, which indicates that all eight factors have an impact on the spatiotemporal evolution characteristics of the CCD of the TEE system in the PRD. (1) The impact of eco-environment protection factors was the largest and has been strengthened, which indicated that the increasing attention paid to eco-environment protection in the past 10 years had a significant role in promoting the CCD of the TEE system in the PRD, especially with the rapid growth of economy as well as urbanization expansion in the later period, resulting in a weakened carrying capacity of the eco-environment. The government attached importance to strengthening the environmental governance cooperation among cities in the PRD to promote harmony between man and nature. (2) The degree of opening-up ranked second. As a vital part of the Guangdong–Hong Kong–Macao Greater Bay Area, the PRD urban agglomeration has attracted a large number of foreign-funded enterprises with its superior geographical location and national preferential policies, which fulfil a leading role in the local economy as well as tourism development. (3) The PRD belongs to the Lingnan cultural zone, which has a profound historical origin and cultural tradition. The PRD is a gathering place of colleges and research institutes. For example, as the center of talent training, scientific research and exchange in South China, Guangzhou Higher Education Mega Center has ten universities. The government has firmly implemented the strategy of “develop the country through science and education”, and has increased investment in education. On the one hand, it helps to cultivate innovative scientific and technological talents and attract high-quality talents, thus promoting industrial upgrading and regional economic development. On the other hand, it helps to narrow regional differences and create a good humanistic environment for regional coordinated development. (4) Technological innovation has a significant positive impact on the CCD of urban agglomeration. The progress of science and technology is not only conducive to promoting industrial upgrading, but also to improving the level of clean technology and green environmental protection technology. In 2006, Guangdong Province approved the several policies of the Guangdong Provincial People’s Government on promoting independent innovation. In 2019, the development planning outline of the Guangdong–Hong Kong–Macao Greater Bay Area required that the PRD region pays attention to eco-environmental protection and the improvement of environmental quality while deeply implementing the strategy of innovation-driven development and vigorously developing technological innovation. (5) The regions with advanced economic development level not only have comprehensive infrastructure, but also have high income of local residents. The economic development of the PRD has driven the development of local tourism, thus promoting coordinated development. (6) The tourism resource endowment affects the level of tourism development. The PRD region takes the opportunity of building the 21st Century Maritime Silk Road and aims to build world-class tourism destination, with diversified tourism resources and rich cultural connotation. (7) There is a positive correlation between the population size factor and the CCD, but its influence gradually weakens, which has a great relationship with the gradual fade of the population dividend in the PRD region. (8) The optimization of industrial structure has a significant positive impact on the CCD, especially increasing the proportion of tertiary industry, which is beneficial to the coordinated development of economy and environment, thereby reducing the dependence on energy, and providing financial support for R&D of environmental protection technologies.

## 4. Discussions and Conclusions

### 4.1. Findings and Discussions

The PRD urban agglomeration is one of the fastest developing regions in China and is an important tourist destination. Using the panel data of nine cities in the PRD from 2010 to 2019, this study tried to analyze the spatial–temporal dynamic characteristics of the coordinated development of the TEE system. The difference from most previous studies is that a more comprehensive and reasonable multi-level indicator system has been established and applied to the study of typical regions of the PRD urban agglomeration. A more in-depth analysis has been conducted from the perspective of time and space, and the dominant factors relating to the CCD has been identified, which enriched the dimension and depth of relevant research. There are similarities and differences between the results of this study and existing studies.

From the perspective of time evolution, the comprehensive benefits of the PRD’s TEE system showed an increasing trend, in which the benefit index of the tourism and economy subsystem increased more rapidly, while the eco-environment subsystem was relatively unstable. Yuan et al. studied the coupling coordination relationship between tourism industry and ecological civilization in Guangdong Province, and found that ecological civilization fluctuated to a certain extent, showing an overall upward trend [56]. However, this study found that the eco-environmental benefits showed a downward trend from 2015 to 2019 in the PRD, indicating that the long-term accumulation of eco-environmental problems in Guangdong Province was more prominent in the PRD urban agglomeration. Lu, Zhang et al. also found in relevant studies in Gansu Province [23] and western China [48] that with the economic growth and tourism expansion, the eco-environment faces risks and development lags behind, indicating that other regions in China also face similar problems in development.

From the view of spatial distribution, the tourism benefits of nine cities in the PRD showed spatial distribution characteristics of high in the east and low in the west, and the regions with the greatest tourism benefits have gradually shifted to the northeast regions of the PRD. The economic benefits presented an uneven but relatively stable spatial pattern, forming a high-value cluster with Guangzhou and Shenzhen at the core, as well as low-value regions in surrounding cities. The eco-environmental benefits showed a fluctuating trend, presenting a spatial pattern of cities with low eco-environmental benefits in the center and cities with high eco-environmental benefits in the surrounding regions. Therefore, there are significant regional variations in the tourism, economic, and eco-environmental benefits of the PRD urban agglomeration, indicating that the PRD has its own spatial distribution characteristics.

That reveals that the CCD of the TEE system in the PRD has been improved with varying degrees for the past decade. The speed of improvement in the early stage was faster than that in the later stage. The lagging types changed from tourism industry to the economy and then to the environment. The coupling coordination type gradually transitioned from one of moderate disorder in the early state to high-quality coordination. Lai, Li et al. have also conducted relevant research on China as a whole or at the regional level [47,49], indicating that the CCD shows an increasing trend or a fluctuating growth trend and generally is developing in the direction of good or high coordination, consistent with this study. Sun et al. analyzed the spatiotemporal evolution trend of coordination between urbanization and eco-environment in the PRD area from 2000 to 2015 [57]. Their research results show that, from the perspective of spatial distribution, the CCD of the west bank of the Pearl River was higher than that of the east bank, and the CDD values of Zhongshan, Zhaoqing, and Guangzhou were the highest. The results of this study showed that Shenzhen, Guangzhou, and Zhuhai had the highest CDD values. From the perspective of spatial change characteristics, the CCD of the TEE system presented a spatial distribution pattern taking the cities near the Pearl River Estuary as the center and decreasing to both sides. Obviously, the two results are not the same, which are related to the two studies using different indicator systems. It can be seen that there is general convergence and individual differentiation in the development of CCD in the PRD.

This study identifies the influencing factors of the CCD through geographical detector analysis, contributing to an in-depth exploring of the driving factors of regional development and providing a decision-making basis for the coming economic structure adjustment, eco-environment protection, and tourism improvement of urban agglomerations [58]. The existing research results have indicated that economic development level, technological innovation level, urbanization, tourism investment, tourism resources, population size, openness, and industrial structure have an impact on CCD [24,55,59], among which technological innovation, economic development, and urbanization are considered to be important factors affecting coordinated development of regional economy and eco-environment. This study found that the spatial–temporal evolution characteristics of the CCD of the TEE system in the PRD were mainly affected by several factors, including eco-environment protection, the degree of opening-up, education investment, technological innovation level, and regional economic development level, which were the result of the interaction of various factors. This is related to the unique development model of the PRD. The rapid development of export-oriented economy by undertaking international industrial transfer has led to increasingly exposed environmental problems in the PRD, so it gradually shifted from low-end manufacturing industry dominated by labor-intensive to high-end manufacturing industry relying on technological innovation, focusing on education and innovation to achieve innovation driven development.

### 4.2. Limitations and Future Work

This research has explored the coupled coordination relationship and its evolution characteristics between the TEE system in the PRD in the past 10 years, which has profound theoretical significance and practical value for the tourism industry, economic growth, and eco-environment optimization of the urban agglomeration. There are still some limitations in this paper. Due to data limitations, the comprehensive Greater Bay Area has not been studied. In order to actively promote the high-quality development of urban agglomeration, the research area can be expanded to Greater Bay Area and even all urban agglomerations in China. Grey system model, multiple regression model, ARIMA-BP combination model, etc., can be used to predict the CCD in the future. In addition, future research can extend the data period to explore the impact of this public health event on regional coordinated development. 

### 4.3. Recommendations

To better realize regional sustainable development, the following recommendations are put forward.

The radiation effect and driving role of primary cities should be fully utilized. The tourism industry and economic growth of the PRD urban agglomeration present primary distribution patterns. The “two-city linkage” of Guangzhou and Shenzhen has shaped the Guangzhou–Shenzhen dual center model, with strong tourism and high economic level. Thus, it is needed to strengthen the central role and driving function of the two cities to drive the tourism and socio-economic development of surrounding cities.Urban circles of sustainable development should be cultivated and built. Guangzhou, Foshan, and Zhaoqing share a common “Guangfu culture.” Cooperation between the cities should be strengthened, giving play to Guangzhou’s function as a driver of the tourism industry and the economic improvement of Foshan and Zhaoqing, relieving the pressure on Guangzhou’s ecological environment, and building the Guangzhou–Foshan–Zhaoqing urban circle into a cooperation demonstration zone in the PRD. Shenzhen, Huizhou, and Dongguan have active investment markets and coastal tourism resources and should rely on national policy support, combine their own unique advantages, deepen industrial reform, and shape diversified markets. Zhuhai, Zhongshan, and Jiangmen should also join forces to strengthen cooperation, improve the transportation network, increase investment in science and technology, and cultivate new drivers of tourism to form a powerful Zhuhai–Zhongshan–Jiangmen growth pole.Tourism development should be driven by the economy. Economic development is the prerequisite for regional tourism development. The relevant authorities should adopt sound fiscal policies, upgrade the industrial structure, develop the tertiary industry, strengthen technological innovation, and strive to practice innovative, green and coordinated sustainable development to inject vitality into local economic development.The ecological concept of “green water and green mountains are golden mountains and silver mountains” should be adhered to. We should fully tap the tourism resources in the western areas of the PRD, develop characteristic tourism projects and ecological recreation projects, build ecotourism demonstration areas, narrow the gap between tourism and socio-economic development in the eastern and western regions of the PRD, and coordinate the relationship between the tourism industry, economic development, and eco-environmental protection.

## Figures and Tables

**Figure 1 ijerph-20-01981-f001:**
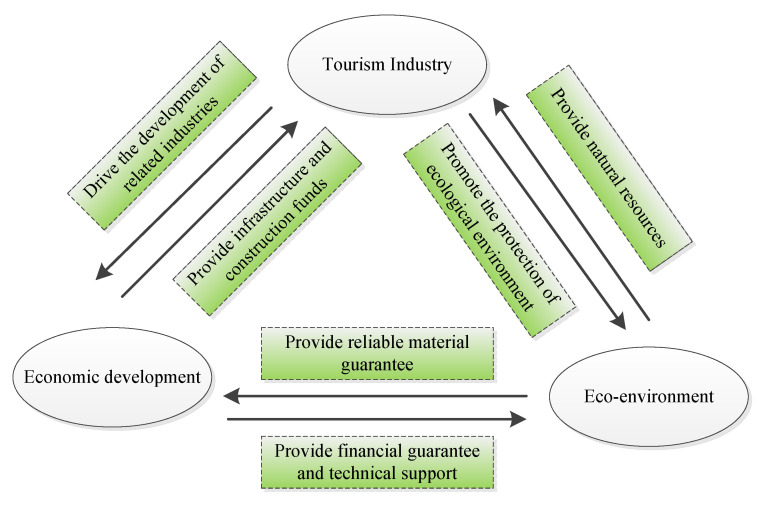
Coupling coordination mechanism of the TEE system.

**Figure 2 ijerph-20-01981-f002:**
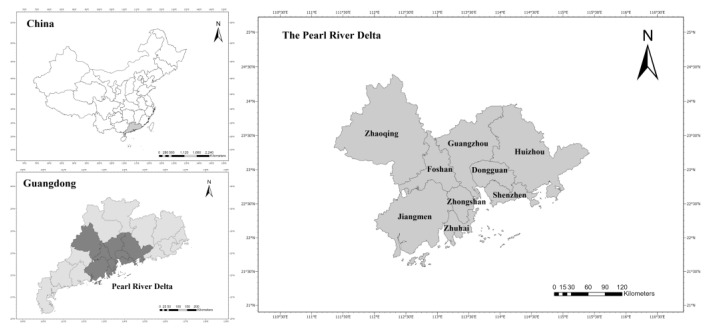
The location of the Pearl River Delta urban agglomeration.

**Figure 3 ijerph-20-01981-f003:**
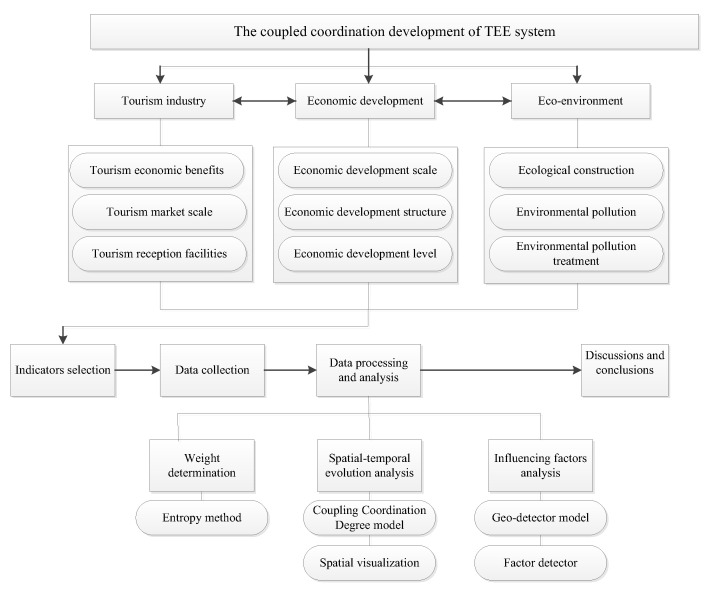
Research framework of the coupled coordination development of the TEE system.

**Figure 4 ijerph-20-01981-f004:**
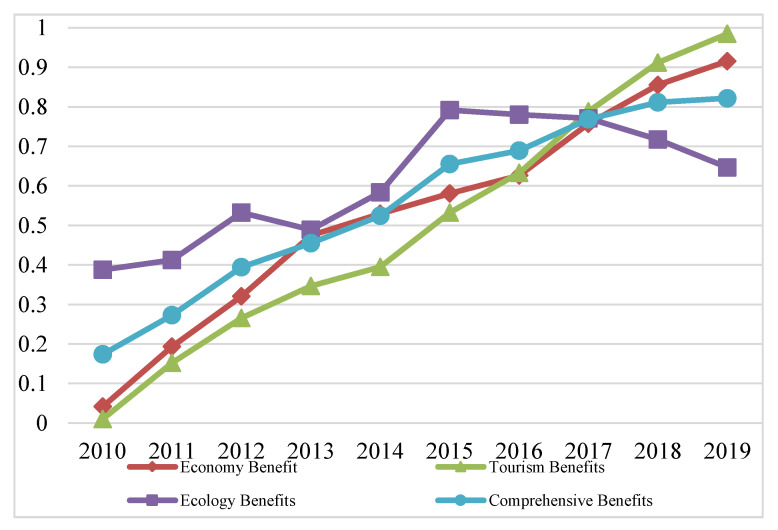
The comprehensive benefits of each subsystem and the TEE system in the PRD.

**Figure 5 ijerph-20-01981-f005:**
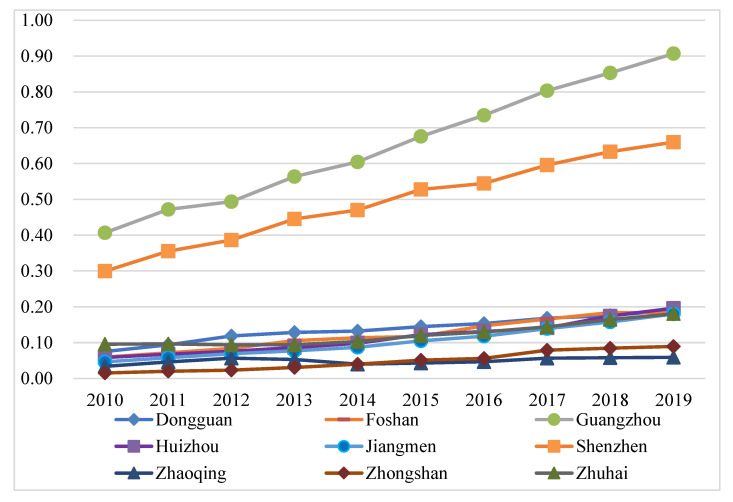
The benefit index of the tourism system in nine cities in the PRD.

**Figure 6 ijerph-20-01981-f006:**
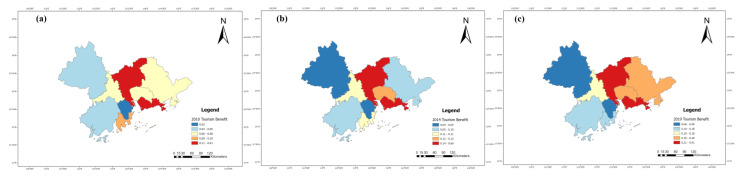
The benefit index of the tourism system in the PRD in (**a**) 2010, (**b**) 2014, and (**c**) 2019.

**Figure 7 ijerph-20-01981-f007:**
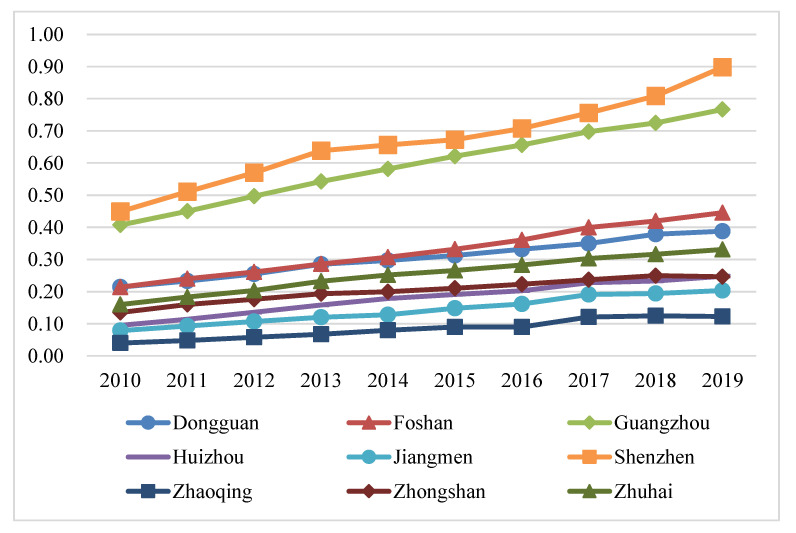
The benefit index of the economic development system in nine cities in the PRD.

**Figure 8 ijerph-20-01981-f008:**
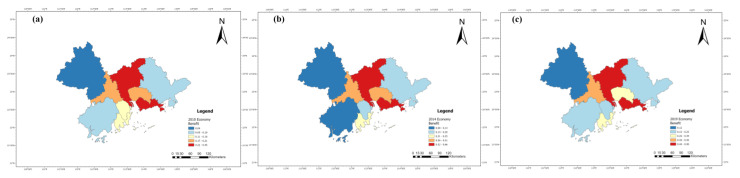
The benefit index of the economic development system in the PRD in (**a**) 2010, (**b**) 2014, and (**c**) 2019.

**Figure 9 ijerph-20-01981-f009:**
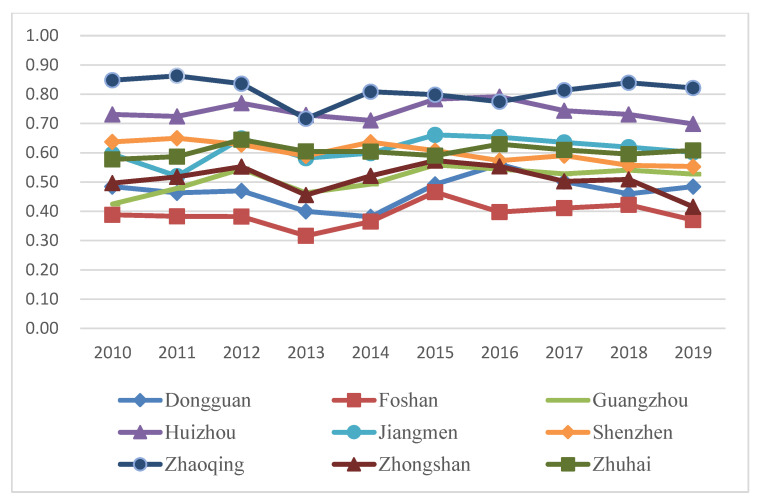
The benefit index of the eco-environment system in nine cities in the PRD.

**Figure 10 ijerph-20-01981-f010:**
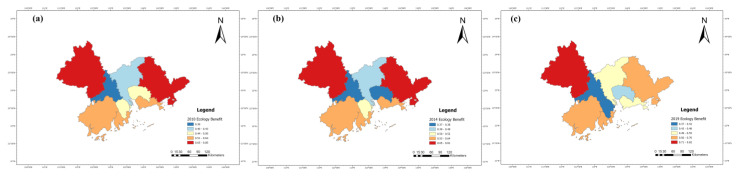
The benefit index of the eco-environment system in the PRD in (**a**) 2010, (**b**) 2014, and (**c**) 2019.

**Figure 11 ijerph-20-01981-f011:**
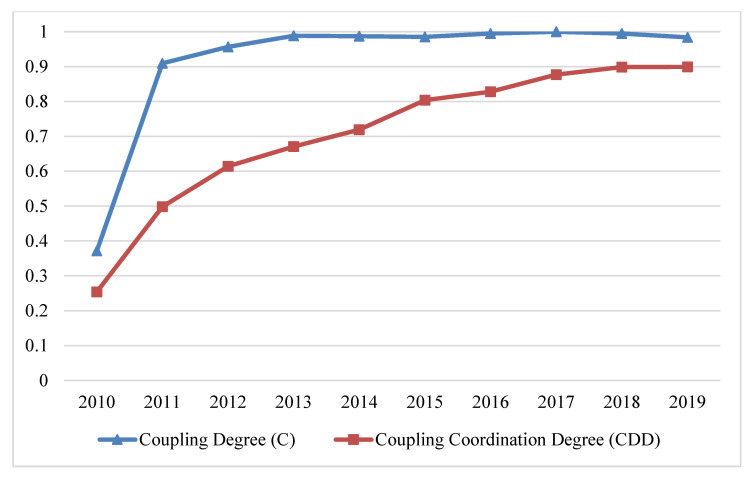
The C and CCD of the TEE system in the PRD.

**Figure 12 ijerph-20-01981-f012:**
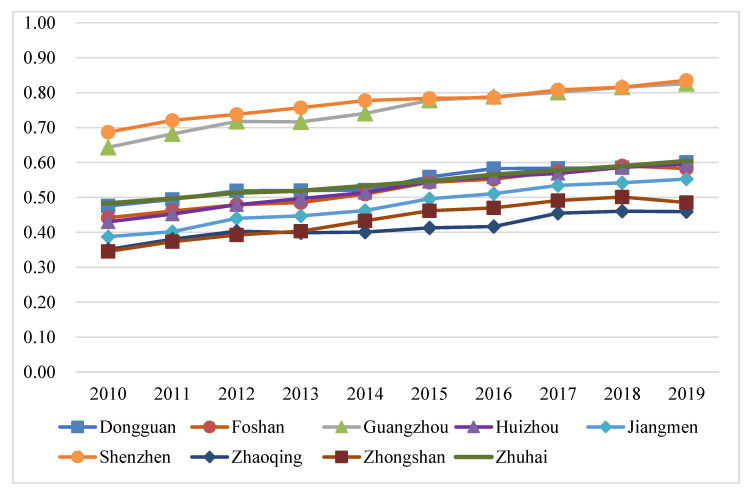
The CCD of the TEE system in nine cities in the PRD.

**Figure 13 ijerph-20-01981-f013:**
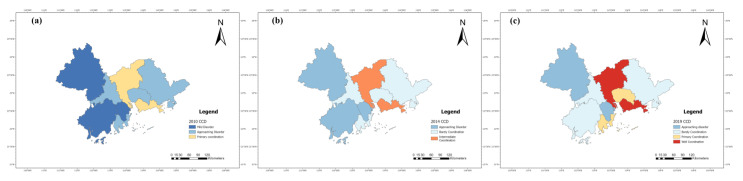
The CCD of the TEE system in the PRD in (**a**) 2010, (**b**) 2014, and (**c**) 2019.

**Table 1 ijerph-20-01981-t001:** The indicator system of three subsystems in the PRD.

System Layer	Primary Indicators	Secondary Indicators	Unit	Type	Weight
Tourism Industry System (X)	Tourism economic benefits X_1_	Total tourism revenue X_11_	RMB 100 million	+	0.1667
		Domestic tourism revenue X_12_	RMB 100 million	+	0.1786
		International tourism revenue X_13_	USD 10,000	+	0.1286
	Tourism market scale X_2_	Number of domestic overnight tourists X_21_	10,000 people	+	0.0911
		Number of inbound overnight tourists X_22_	10,000 people	+	0.1450
	Tourism reception facilities X_3_	Number of A-level scenic spots X_31_	a	+	0.1060
		Number of travel agencies X_32_	a	+	0.1840
Economic Development System (Y)	Economic development scale Y_1_	GDP Y_11_	RMB 100 million	+	0.1664
		Total retail sales of social consumer goods Y_12_	RMB 100 million	+	0.1629
		Total export-import volume Y_13_	USD 10,000	+	0.1816
		Fixed asset investment in the whole society Y_14_	RMB 100 million	+	0.1196
	Economic development structure Y_2_	Proportion of primary industry in GDP Y_21_	%	−	0.0283
		Proportion of second industry in GDP Y_22_	%	+	0.0317
		Proportion of tertiary industry in GDP Y_23_	%	+	0.0921
	Economic development level Y_3_	GDP per capita Y_31_	RMB/people	+	0.0532
		Per capita disposable income Y_32_	RMB/people	+	0.0618
		Per capita local government general public budget revenue Y_33_	RMB/people	+	0.1024
Eco-environment System (Z)	Ecological construction Z_1_	Per capita park green area Z_11_	hm^2^/person	+	0.1391
		Green coverage rate in built-up areas Z_12_	%	+	0.0302
		Forest coverage Z_13_	%	+	0.2928
		Percentage of days with good air quality Z_14_	%	+	0.1128
	Environmental pollution Z_2_	Industrial solid wastes produced Z_21_	10,000 tons	−	0.0679
		Industrial wastewater discharge Z_22_	10,000 tons	−	0.1076
		Industrial waste gas emission Z_23_	100 million m^3^	−	0.1229
	Environmental pollution treatment Z_3_	Centralized treatment rate of wastewater treatment plants Z_31_	%	+	0.0329
		Pollution-free treatment rate of domestic garbage Z_32_	%	+	0.0353
		Comprehensive utilization rate of industrial solid waste Z_33_	%	+	0.0586

Note: “+” indicates a positive indicator; “−” indicates a negative indicator.

**Table 2 ijerph-20-01981-t002:** The classification criteria of CCD of the TTE system.

Category	CCD	Coordination State	Category	CCD	Coordination State
Disorder	0.00–0.09	Extreme disorder	Coordination	0.5–0.59	Bare coordination
	0.1–0.19	Severe disorder		0.6–0.69	Primary coordination
	0.2–0.29	Moderate disorder		0.7–0.79	Intermediate coordination
	0.3–0.39	Mild disorder		0.8–0.89	Well coordination
	0.4–0.49	Approaching disorder		0.9–1.00	High-quality coordination

**Table 3 ijerph-20-01981-t003:** The influencing factors of CCD of the TEE system in PDR.

Influencing Factors	Indicators	Code
Degree of opening-up	Total value of actual utilized foreign capital/USD 10,000	X1
Education investment	Proportion of education expenditure in fiscal expenditure/%	X2
Regional economic development level	GDP growth rate/%	X3
Industrial structure	Proportion of second and tertiary industry in GDP/%	X4
Tourism resource endowment	Number of A-level scenic spots/a	X5
Eco-environment protection	Total capacity of sewage treatment/10,000 m^3^	X6
population size	Year-end population/10,000 people	X7
Technological innovation level	Number of ordinary colleges and universities/unit	X8

**Table 4 ijerph-20-01981-t004:** Comprehensive evaluation values and classification types of CCD of the TEE system in the PDR.

Year	C	T	D	CCD Types	F(x)	G(y)	H(z)	F(x), G(y), H(z)	Lag Types
2010	0.371	0.174	0.254	Moderate disorder	0.01	0.041	0.388	F(x) < G(y), H(z)	Tourism lagging
2011	0.909	0.273	0.498	Bare coordination	0.152	0.193	0.413	F(x) < G(y), H(z)	Tourism lagging
2012	0.956	0.394	0.614	Primary coordination	0.266	0.321	0.532	F(x) < G(y), H(z)	Tourism lagging
2013	0.988	0.455	0.670	Primary coordination	0.346	0.475	0.489	F(x) < G(y), H(z)	Tourism lagging
2014	0.987	0.524	0.719	Intermediate coordination	0.395	0.530	0.584	F(x) < G(y), H(z)	Tourism lagging
2015	0.985	0.656	0.804	Well coordination	0.533	0.581	0.792	F(x) < G(y), H(z)	Tourism lagging
2016	0.995	0.689	0.828	Well coordination	0.633	0.626	0.780	G(y) < F(x), H(z)	Economic lagging
2017	0.999	0.769	0.877	Well coordination	0.788	0.757	0.771	G(y) < H(z), F(x)	Economic lagging
2018	0.995	0.812	0.899	Well coordination	0.911	0.856	0.717	H(z) < G(y), F(x)	Environment lagging
2019	0.984	0.822	0.9	High-quality coordination	0.985	0.916	0.647	H(z) < G(y), F(x)	Environment lagging

**Table 5 ijerph-20-01981-t005:** The impact of various indicators on the CCD of the TEE system in PDR.

Code	Influencing Factors	2010		2014		2019		q-Mean
		q-Value	*p* Value	q-Value	*p* Value	q-Value	*p* Value	
X1	Degree of opening-up	0.902	0.000	0.884	0.000	0.971	0.000	0.919
X2	Education investment	0.975	0.000	0.874	0.000	0.644	0.000	0.831
X3	Regional economic development level	0.842	0.000	0.874	0.000	0.748	0.000	0.821
X4	Industrial structure	0.704	0.000	0.603	0.000	0.719	0.000	0.675
X5	Tourism resource endowment	0.699	0.000	0.870	0.000	0.809	0.000	0.793
X6	Eco-environment protection	0.896	0.000	0.973	0.000	0.998	0.000	0.956
X7	population size	0.885	0.000	0.767	0.000	0.639	0.000	0.764
X8	Technological innovation level	0.870	0.000	0.869	0.000	0.734	0.000	0.824

Note: All *p* values are less than 0.001, indicating that the factor explanatory power q values are all significant at the 99% level.

## Data Availability

The data presented in this study are available on request from the corresponding author. The data are not publicly available due to data publisher regulations.

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
