# Peer review of "Exploring the Coordination and Spatial–Temporal Characteristics of the Tourism–Economy–Environment Development in the Pearl River Delta Urban Agglomeration, China"

_ijerph, 2023, doi:10.3390/ijerph20031981_

Round 1

Reviewer 1 Report

This paper aims to develop a comprehensive indicator system to evaluate the coordination of the tourism-economy-environment (TEE) system based on a case study of the Pearl River Delta (PRD). The paper is well-written and rich in information. However, I have a few questions/suggestions for the authors to consider.

First, the theoretical contribution of this study needs to be emphasized. While the authors claim that “few studies take urban agglomeration as the research object to analyze the spatial-temporal characteristics and influencing factors of the CCD”, it is not clear why urban agglomeration is an important spatial unit for relevant research. The authors did not discuss what new perspectives or findings are added through this special case study, which should be added at least in the conclusion.

 Second, the choice of period (2010 to 2019) should also be explained. It has already been three years after 2019, why new data are not included in the analysis?

 Third, the analytical part is too descriptive. The authors only briefly present the findings of analysis, without discussing their theoretical implications. Additionally, some explanations of the results should also be added.

 Fourth, the analysis of the influencing factors of CCD should also be further strengthened. Instead of merely presenting the outcomes, more explanations are also needed.

Fifth, some figures could be merged to make the paper more compact.

Author Response

Dear reviewer,

We would like to thank you for your time and effort in reviewing our paper and providing constructive comments. Those comments are all valuable and very helpful for revising and improving our manuscript, as well as the important guiding significance to our researches. We have studied the comments carefully and the manuscript has been revised carefully and strictly according to your comments, which we hope meet with approval.

We are here resubmitting the revised manuscript entitled “Exploring the Coordination and Spatial-Temporal Characteristics of the Tourism-Economy-Environment Development in the Pearl River Delta Urban Agglomeration, China” for your kind consideration of its suitability for publication in International Journal of Environmental Research and Public Health. In order to facilitate your review, bold and marked red fonts were used to show the corresponding response. And we have also updated our manuscript by properly adding these responses into the revised version.

Our deepest gratitude goes to you for your careful work and thoughtful suggestions that have helped improve this paper substantially.Please see the attachment.

Best Regards,

Authors

Reviewer 2 Report

i am honor to review this paper, i have the following suggestions:

(1) In terms of your title, you better refine a better title, this title is so common and featureless.

(2)An academic article should clearly reflect on the research questions it asks and their context. why do you study this issue, why this issue is special in Pearl River Delta Urban Agglomeration (why not the Yangzte River Delta)

(3) in terms of the indicator system,  why the Proportion of primary industry in GDP is a negative indicator, in therms of the influencing factors, why select these eight indicators, how they can affect the coupling coordination degree.  As you say "with the increasing education investment in the PRD, various scientific research platforms have received a large amount of scientific research funding support, attracted much top talent from China and internationally, improved the independent innovation ability, and optimized industrial structure, enhancing the CCD in the PRD", what is the relatioonship between increasing education investment and the CCD?

(4) There is no discussion of your research resutls, how it compares to previous research, and in what ways it contributes to theoretical and practical significance. There needs brief mentions of the theoretical contributions.

(5) Limited ideas for future research are offered based on the findings of the paper.

Author Response

(The authors gave the same response as above.)

Round 2

Reviewer 1 Report

No further comments.

Author Response

Thank you for your constructive comments and recognition.

All my best wishes for the future.

Reviewer 2 Report

Congraduations!

Author Response

(The authors gave the same response as above.)
